



# Drought and radiation explain swings in Amazon rainforest greenness during the 2015–2016 drought

Yi Y. Liu[1], Albert I. J. M. van Dijk[2], Patrick Meir[3,4], Tim R. McVicar[5]

[1]School of Civil and Environmental Engineering, University of New South Wales, Sydney, New South Wales, 2052, Australia

[2]Fenner School of Environment & Society, Australian National University, Canberra, Australian Capital Territory, 0200, Australia

[3]Research School of Biology, Australian National University, Canberra, Australian Capital Territory, 0200, Australia

[4]School of Geosciences, University of Edinburgh, Alexander Crum Brown Road, Edinburgh, EH93FF, UK

[5]CSIRO Environment, GPO Box 1700, Canberra, Australian Capital Territory, 2601, Australia

*Correspondence to*: Yi Y. Liu (yi.liu@unsw.edu.au)

**Abstract.** The 2015/16 Amazon drought was characterized by below-average regional precipitation for an entire year, which distinguishes it from the dry-season only droughts in 2005 and 2010. Studies of vegetation indices (*VI*) derived from optical remote sensing over the Amazonian forests indicated three stages in canopy response during the 2015/16 drought, with below-average greenness during the onset and end of the drought, and above-average greenness during the intervening months. So far, a satisfactory explanation for this broad temporal pattern, and spatial variation within the Amazonian forests of this broad response, has not been found. Better understanding of rainforest behaviors during this unusually long drought should help predict their response to future droughts. We hypothesized that below-average greenness could be explained by water deficit and heat stress occurring beyond the tolerance thresholds of rainforest. To test our hypothesis, we used monthly observations of terrestrial water storage (*TWS*), land surface temperature (*LST*) and vapor pressure deficit (*VPD*) for January 2003–December 2016. First, for each 1° grid cell, we determined the 'normal' range of monthly *TWS*, *LST* and *VPD* during non-drought years (i.e. 2003–2016, excluding 2005, 2010, 2015 and 2016), and identified the extreme values of 'normal' range, i.e. minimum *TWS*, maximum *LST* and maximum *VPD*. We considered the normal hydrological and thermal ranges to have been exceeded when (1) two or three of these variables were simultaneously beyond their extreme values, or (2) only one variable was beyond the extreme value, but the other two were significantly ($p<0.05$) different from the average for non-drought years. Using these criteria, regions experiencing hydrological and thermal conditions beyond the 'normal' range during different stages of the 2015/16 event were delineated. The results showed a gradual southward shift of these regions: from the north-eastern Amazon in August–October 2015, to the north-central part in November 2015–February 2016 and finally to the southern Amazon in July 2016. The majority of forests within the delimited regions experienced below-average greenness. Conversely, outside of these regions, greenness responded positively to radiation anomalies, as is expected under



normal conditions. The opposing influences of drought and radiation anomalies together explained more than 70% of the observed spatiotemporal patterns in greenness. These results suggest that our exceeding 'normal' ranges based approach, combining water storage, temperature and atmospheric moisture demand drivers, can reasonably identify the most likely drought-affected regions at monthly to seasonal time scales. Using observation-based hydrological and thermal condition

thresholds can help with interpreting the response of Amazon rainforest to future drought events.

## 1 Introduction

The Amazon rainforest is the largest contiguous area of tropical rainforest in the world and plays a crucial role in the water cycle and carbon budget, both regionally and globally (Tian et al., 1998; Pan et al., 2011; Ahlström et al., 2015). In little more than one decade, three record-breaking droughts have hit the region in 2005, 2010 (Marengo and Espinoza, 2016) and

2015/16 (Jiménez-Muñoz et al., 2016). Hydro-meteorological signals observed in the 2005 and 2010 droughts include a strong precipitation deficit during the extended dry season (Liu et al., 2018), low river discharge and total water storage (Xu et al., 2011), high canopy temperatures (Toomey et al., 2011) and enhanced atmospheric moisture demand (Lee et al., 2013). These resulted in widespread reductions in canopy photosynthesis and canopy water content (Xu et al., 2011; Saatchi et al., 2012; Lee et al., 2013; Liu et al., 2018), a slowdown of forest growth, and increased tree mortality (Phillips et al., 2009;

Lewis et al., 2011; Gatti et al., 2014; Feldpausch et al., 2016; Hubau et al., 2020).

The 2005 and 2010 droughts occurred primarily during the extended dry season, from May through October (Liu et al., 2018). In contrast, during the 2015/16 drought below-average regional precipitation and above-average radiation occurred for a full year, from August 2015 through July 2016, i.e. from the dry season of 2015 to the dry season of 2016 (Yang et al.,

2018). The 2015/16 drought was also characterized by high temperatures (Yue et al., 2017) and low water storage (Erfanian et al., 2017). Long- and short-term responses to drought by tropical forests may differ in key respects (Meir et al., 2018). An analysis of Amazon forest response during the unusually prolonged drought of 2015/16, in comparison to previous, shorter droughts, may provide new insights into the underlying mechanisms and help predict forest response in a changing climate at monthly to inter-annual timescales.


Two vegetation indices, Normalized Difference Vegetation Index (NDVI) and the Enhanced Vegetation Index (EVI), have been derived from the optical Moderate Resolution Imaging Spectroradiometer (MODIS) instruments on NASA's Terra and Aqua satellites and are the most commonly used data to characterize Amazon rainforest canopy dynamics (Xiao et al., 2006; Anderson et al., 2010; Atkinson et al., 2011; Galvao et al., 2011; Samanta et al., 2012; Hilker et al., 2015; Maeda et al.,

2016). Both vegetation indices (*VI*) provide measures of canopy 'greenness' that have been shown to correlate well to canopy photosynthetic capacity, which itself is the combined result of leaf chlorophyll, leaf age, canopy cover and structure (Ramachandran et al., 2011). While the NDVI is sensitive to chlorophyll abundance, the EVI is more responsive to canopy





structural variations, and the two indices are to some degree complementary in detecting vegetation change (Huete et al., 2002). An important feature of MODIS *VI* is that they capture widespread canopy greening in response to increased solar

radiation during the dry season of non-drought years (Huete et al., 2006). This phenological response has been confirmed by field measurements (Restrepo-Coupe et al., 2013; Saleska et al., 2016; Wu et al., 2018, Gonçalves et al., 2023).

Previous studies used MODIS *VI* to examine the dynamics of Amazon rainforest greenness during the 2015/16 drought (Yang et al., 2018; Yan et al., 2019). Over the 12-month period August 2015–July 2016, the spatial patterns of greenness and

radiation anomalies were positively correlated (Yang et al., 2018) (Fig. 1a and b). However, at shorter time scales, the agreement breaks down (Fig. 1c-l). Regional greenness appeared below average at the start (August–October 2015) and end (July 2016) of the 12-month drought, but above or close to average during the intervening eight months (Fig. 1c). This temporal pattern was also found by Yang et al. (2018) and Yan et al. (2019), despite slight differences in the *VI*s products used and study periods. The 12 months (i.e. August 2015–July 2016) can be divided into four stages according to greenness

anomaly: (Stage I) below average during August–October 2015, (Stage II) close to average during November 2015–February 2016, (Stage III) above average during March–June 2016 and (Stage IV) below average in July 2016. Meanwhile, radiation remained above average for most of the 2015/16 event, though it was close to average during Stage III (Fig. 1d). Spatially, the discrepancy between the anomalies in greenness and radiation was the most striking in Stages I and IV, i.e. below-average greenness but above-average radiation over the northeast during August–October 2015 (Fig. 1e and f) and south in

July 2016 (Fig. 1k and l). This discrepancy suggests that other factors, in addition to radiation, played a role in controlling greenness in the first and last months of the 2015/16 drought event. Several potential driving factors could be expected to be correlated, including radiation, moisture availability and temperature. This makes it challenging to identify their individual contributions. Better understanding of their interactions during the 2015/16 drought should help improve our capacity to predict canopy responses to future droughts, which may become more frequent, severe and/or longer (Malhi et al., 2008;

Meir and Woodware, 2010).





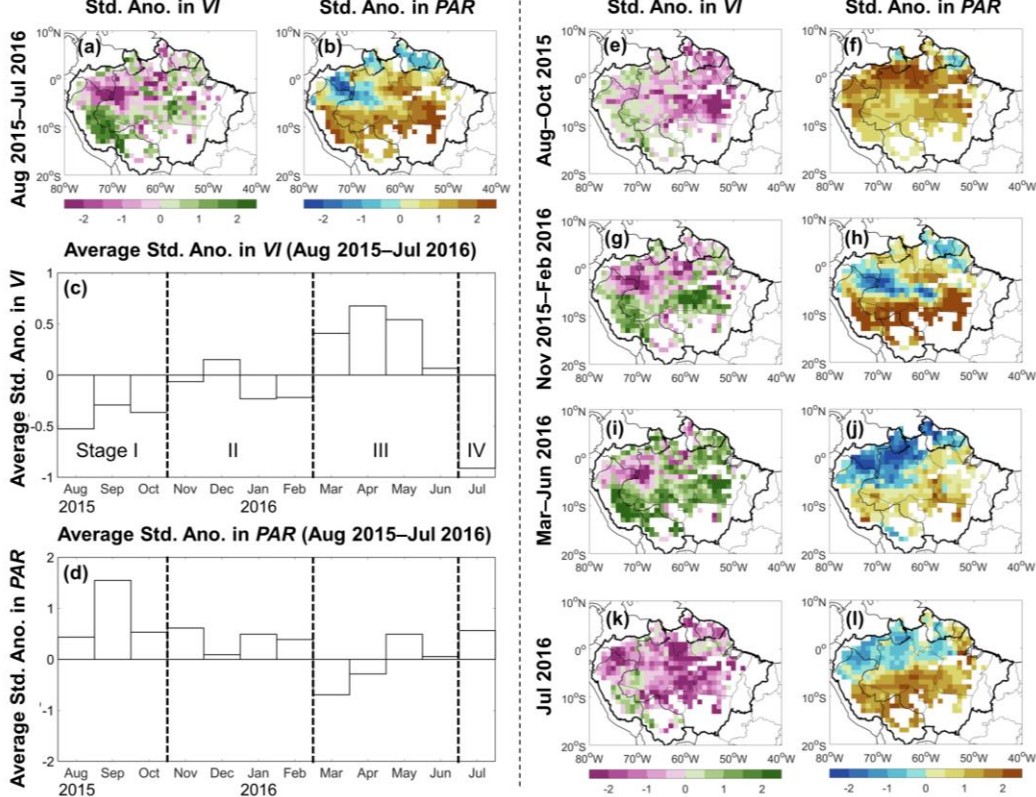

**Figure 1.** Standardized anomalies (Std. Ano.) in vegetation indices (*VI*) and photosynthetically active radiation (*PAR*) during the 2015/16 Amazon drought over the 1° grid cells with more than 80% covered by 'evergreen broadleaf forests'. Panels (a) and (b) are the spatial distribution of standardized anomalies in *VI* and *PAR* for the 12 months between August 2015 and July 2016, respectively. Units measure how many standard deviations from the non-drought years' average (i.e. 2003–2016, excluding four drought years 2005, 2010, 2015 and 2016). Standardized anomaly in EVI was calculated for each grid cell first; the same for NDVI. We took the mean value of these two standardized anomalies and considered it as the standardized anomaly in *VI*, as EVI and NDVI provide complementary information to each other (Huete et al. 2002). Panels (c) and (d) show the regional average standardized anomaly in *VI* and *PAR* for each month from August 2015 through July 2016. These 12 months can be divided into four stages based on the anomaly directions of *VI*. Panels (e) to (l) are the spatial distribution of standardized anomalies in *VI* and *PAR*, for each of the four stages defined in panel (c). More details about data sources and pre-processing of *VI* and *PAR* can be found Table 1 and the Methods section, respectively.

Interpretation of EVI and NDVI over the Amazon rainforest has been challenging as their temporal variation is small and influenced by sun-target-sensor geometry changes as well as clouds and aerosols (Samanta et al., 2010; Morton et al., 2014; Saleska et al., 2016). Based on EVI and NDVI derived from the MODIS, widespread below-average greenness was observed in the dry season (July–September) during the 2010 Amazon drought (Atkinson et al., 2011; Xu et al., 2011). However, using the same data, there has been debate around greenness anomalies in the dry season of 2005 drought (Saleska et al., 2007; Samanta et al., 2010). Considerable efforts have been made to apply more accurate atmospheric correction, cloud detection, improved sensor calibration and sun-target-sensor geometry correction (Lyapustin et al., 2011a; Lyapustin et al., 2011b; Lyapustin et al., 2012), but some noise may still persist (Bi et al., 2016; Maeda et al., 2016). In addition to vegetation





observations, independent satellite observations of, among others, precipitation, temperature and terrestrial water storage are also available since around 2000. This provides an opportunity to draw on multiple lines of evidence and characterize the hydro-meteorological drivers of rainforest response. Spatiotemporal consistency among these independent observations may increase the certainty of interpretation thus indicating the most likely eco-hydrological mechanisms involved.

Field experiments suggest that the Amazon rainforest has water and heat threshold limits beyond which normal physiological behavior is adversely affected (Meir et al., 2015). In the dry season of non-drought years, soil water is found sufficient for both sap flow and transpiration to occur even when soil water content reaches its annual minimum value (Fisher et al., 2006; Fisher et al., 2007; Nepstad et al., 2007; Meir et al., 2009; Wu et al., 2016; da Costa et al., 2018; Meir et al., 2018; Meng et al., 2022). This indicates that the soil profile can supply enough water during a normal dry season, probably assisted by

deeper root systems (Nepstad et al., 1994; Yang et al., 2016). However, when the dry season coincides with a drought, there can be a limit to this capacity. For example, in an experiment preventing 50% of precipitation falling through the canopy from infiltrating into the soil, soil water availability was apparently below the minimum for non-drought years (Meir et al., 2015). As a result, sap flow was reduced considerably (Fisher et al., 2007; da Costa et al., 2018). In addition, there appear to be similar thresholds in canopy temperature and vapor pressure deficit (*VPD*, a measure of atmospheric moisture demand)

(Tan et al., 2017; Pau et al., 2018; Grossiord et al., 2019). Photosynthesis and sap flow rate thus tend to increase with temperature and *VPD* while these remain below the threshold, but decrease beyond it. In non-drought years, Amazon rainforests experience maximum temperature and *VPD* during the dry season (Hutyra et al., 2007). At the same time, new leaf flush occurs and ecosystem photosynthesis can be maintained or increased if dry-season radiation is high and soil moisture supply is sufficient (Carswell et al., 2002).

Accordingly, we hypothesized that the below-average greenness during the 2015/16 drought year was most likely caused by an exceedance of moisture deficit and/or heat tolerance limits, particularly in Stages I and IV. To test our hypothesis, we used data on terrestrial water storage (*TWS*), land surface temperature (*LST*) and vapor pressure deficit (*VPD*) for 2003–2016, which includes both drought and non-drought years. We identified the range of *TWS*, *LST* and *VPD* averaged during

non-drought years (i.e. defined as 2003-2016 excluding four drought years 2005, 2010, 2015 and 2016) for each grid cell, and used these as an estimate of the normal hydrological and thermal range. Subsequently, we mapped when and where this 'normal' range was exceeded during the 2015/16 drought. By comparing their spatiotemporal patterns with those in radiation and greenness anomalies, we sought to explain observed differences in greenness response during the event.





## 2 Data

### 2.1 Data sources

Several eco-hydrological variables were used to characterize the spatiotemporal patterns of greenness and drought during the 2015/16 event (Table 1). They include: (*i*) greenness represented by Enhanced Vegetation Index (EVI) (Huete et al., 1994; Huete et al., 1997) and Normalized Difference Vegetation Index (NDVI) (Tucker, 1979) from the MODIS instrument onboard Aqua (Didan 2015); (*ii*) photosynthetically active radiation (*PAR*, W m$^{-2}$) from the Clouds and Earth's Radiant Energy System (CERES, SYN1deg_Ed4.1) onboard Aqua and Terra (Wielicki et al., 1996); (*iii*) precipitation (*P*, mm month$^{-1}$) derived from the Tropical Rainfall Measuring Mission (TRMM 3B43 v7) (Huffman et al., 2007); (iv) terrestrial water storage (*TWS*, mm) from the Gravity Recovery and Climate Experiment (GRACE Mascons) (Watkins et al., 2015; Wiese et al., 2016; Save et al., 2016; Loomis et al., 2019); (v) land surface temperature (*LST*, K) from the daytime overpasses (1:30 PM) of the Atmospheric Infrared Sounder (AIRS) onboard Aqua (version 7) (Kahn et al., 2014; Susskind et al., 2014; Ding et al., 2020); and (vi) 2 m dewpoint temperature ($T_{dew}$, K) and 2 m temperature ($T_{air}$, K) obtained from the ERA5-Land reanalysis (Copernicus Climate Change Service, 2019) which were used to calculate the atmospheric vapor pressure deficit (*VPD*, kPa).

**Table 1**. Major characteristics of the datasets used herein for January 2003–December 2016.

| Variable | Sources | Original spatial & temporal resolution | Download links (last accessed: 4 September 2023) |
|---|---|---|---|
| Vegetation Indices (*VI*) | MODIS/ Aqua | 0.05°/ monthly | https://e4ftl01.cr.usgs.gov/MOLA/MYD13C2.061 |
| Photosynthetically Active Radiation (*PAR*) | CERES/ Terra and Aqua | 1°/ monthly | https://ceres-tool.larc.nasa.gov/ord-tool/jsp/SYN1degEd41Selection.jsp ('PAR Surface Flux Direct' and 'PAR Surface Flux Diffuse') |
| Precipitation (*P*) | TRMM and other satellites | 0.25°/ monthly | https://disc2.gesdisc.eosdis.nasa.gov/data/TRMM_L3/TRMM_3B43.7 (TRMM 3B43 v7) |
| Terrestrial Water Storage (*TWS*) | GRACE | 0.25° to 1°/ monthly | http://grace.jpl.nasa.gov http://www2.csr.utexas.edu/grace https://earth.gsfc.nasa.gov/geo/data/grace-mascons (Simple arithmetic mean of JPL, CSR and GSFC fields used) |
| Land Surface Temperature (*LST*) | AIRS/ Aqua | 1°/ monthly | https://acdisc.gesdisc.eosdis.nasa.gov/data/Aqua_AIRS_Level3 ('SurfSkinTemp_A') |
| Surface Dewpoint Temperature ($T_{dew}$) and Surface Air Temperature ($T_{air}$) | ERA5-Land | 0.1°/ monthly | https://cds.climate.copernicus.eu/cdsapp#!/dataset/reanalysis-era5-land-monthly-means?tab=form (Product type: Monthly averaged reanalysis; Variables: '2m dewpoint temperature' and '2m temperature') |





**2.2 Data pre-processing**

All data were available at monthly temporal resolution for January 2003–December 2016. All datasets have full 168-month coverage except $TWS$. Occasional months (21 out of 168 months during 2003–2016, the longest gap being three consecutive months) were missing in the original $TWS$ dataset. Missing $TWS$ data are commonly filled using linear interpolation (Chen et al., 2013; Solander et al., 2017), on the assumption that missing data were not local maxima or minima. To avoid this assumption, instead, we gap-filled the missing values by considering their correlation to precipitation and radiation (see Appendix A for details).

Vapor pressure deficit ($VPD$, kPa) is the difference between the vapor pressure when the air is saturated ($e_s$) and actual vapor pressure ($e_a$). Here, $VPD$ was calculated as $e_s$–$e_a$ with the availability of surface dewpoint temperature ($T_{dew}$, °C) and surface air temperature ($T_{air}$, °C) from ERA5-Land reanalysis.

$$e_s = 0.6108 \times \exp((17.27 \times T_{air})/(T_{air}+237.3)) \quad (1)$$

$$e_a = 0.6108 \times \exp((17.27 \times T_{dew})/(T_{dew}+237.3)) \quad (2)$$

To allow direct comparison, all datasets were resampled to 1° resolution by aggregation. The spatial extent of Amazon rainforest was delineated based on the 0.05° MODIS land cover type product (MCD12C1.006) for 2015. To minimize the influence of non-forest vegetation signals, our analysis was limited to 1° grid cells with more than 80% of 0.05° grid cells classified as 'evergreen broadleaf forests' following the International Geosphere-Biosphere Programme (IGBP) classification (Friedl et al., 2010).

**3 Methods**

The analysis approach consisted of three steps.

_Step 1._ We identified the average and extreme values of the three variables ($TWS$, $LST$ and $VPD$) during non-drought years (i.e. all years during 2003–2016 excluding four drought years 2005, 2010, 2015 and 2016) for every grid cell. A detailed example is shown in Fig. 2. For each grid cell, one average value of normal years was calculated for each month from January to December, producing 12 values for each variable (i.e. $TWS_{ND-Ave}$, $LST_{ND-Ave}$ and $VPD_{ND-Ave}$) for each grid cell. The lowest or highest (depending on which indicates water and thermal conditions) among the 12 values was then determined, producing $TWS_{ND-Min}$, $LST_{ND-Max}$, and $VPD_{ND-Max}$. Applying this procedure to all grid cells, we derived twelve maps each of $TWS_{ND-Ave}$, $LST_{ND-Ave}$ and $VPD_{ND-Ave}$ and one map of each $TWS_{ND-Min}$, $LST_{ND-Max}$, and $VPD_{ND-Max}$.





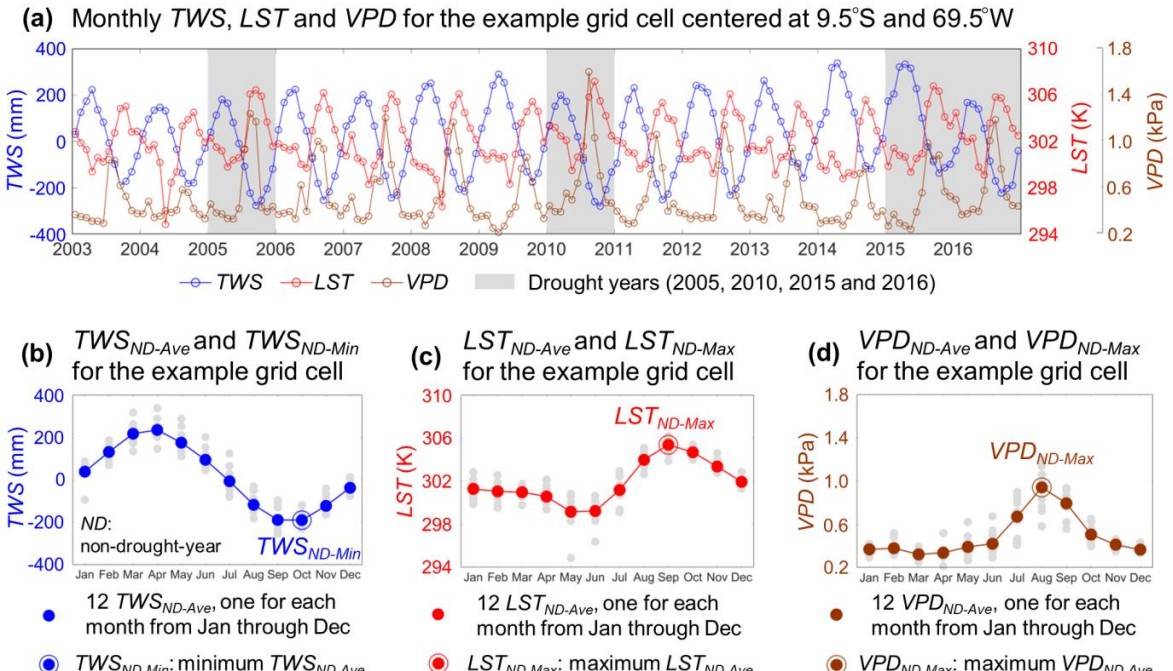

**Figure 2.** Example illustrating how to derive the average and extreme values of *TWS*, *LST* and *VPD* in non-drought years. Panel (a) shows the time series of monthly *TWS*, *LST* and *VPD* for the 1° grid cell centered at 9.5°S, 69.5°W from January 2003 through December 2016. During these 14-years, four years were considered as drought years, i.e. 2005, 2010, 2015 and 2016, while the remaining ten years were deemed as non-drought years. Panel (b) shows how we derived non-drought years' (*ND*) average and minimum *TWS* values, herein denoted as $TWS_{ND\text{-}Ave}$ and $TWS_{ND\text{-}Min}$, respectively. There are 12 $TWS_{ND\text{-}Ave}$ values, one for each month from January to December. For example, $TWS_{ND\text{-}Ave}$ for January is the average *TWS* of all January values of ten non-drought years (grey dots). The minimum value of 12 $TWS_{ND\text{-}Ave}$ was taken as the $TWS_{ND\text{-}Min}$; for this example grid cell, $TWS_{ND\text{-}Min}$ is the October's $TWS_{ND\text{-}Ave}$. Panels (c) and (d) show the same as (b), but for *LST* and *VPD*. The extreme values of *LST* and *VPD* are $LST_{ND\text{-}Max}$ and $VPD_{ND\text{-}Max}$, respectively, which were reached in September and August during non-drought years for this example grid cell.

*Step 2.* We considered that the range of hydrological and thermal conditions experienced during non-drought years as the 'normal' range. To determine whether this 'normal' range was exceeded during 2015/16, we used two criteria: (1) whether exceeding the extreme values, i.e. $TWS < TWS_{ND\text{-}Min}$, $LST > LST_{ND\text{-}Max}$ and $VPD > VPD_{ND\text{-}Max}$; and (2) whether each of the variables showing a statistically significant ($p < 0.05$) deviation from the average for that month in non-drought years, i.e. $TWS < TWS_{ND\text{-}Ave}$ ($p < 0.05$), $LST > LST_{ND\text{-}Ave}$ ($p < 0.05$) and $VPD > VPD_{ND\text{-}Ave}$ ($p < 0.05$). The non-parametric Wilcoxon signed rank test was used to determine the significance level (Gibbons and Chakraborti, 2011). To reduce the uncertainties associated with each of these three variables (i.e. *TWS*, *LST* and *VPD*), a combination of them was used. We considered the 'normal' ranges were exceeded, when: (1) any two or all three of $TWS_{ND\text{-}Min}$, $LST_{ND\text{-}Max}$ and $VPD_{ND\text{-}Max}$ for a grid-cell were exceeded simultaneously; or (2) only one of the three extreme values was exceeded, while the other two variables being significantly departed from their non-drought years' average. We calculated the two criteria for each grid cell for each month from August 2015 through July 2016, and subsequently delineated the regions where the 'normal' ranges were exceeded.





200

*Step 3.* We examined whether incorporating the exceedance of the 'normal' ranges explained the observed greenness, particularly in Stages I and IV (Fig. 1). To achieve this, we first calculated the standardized anomalies in *VI* and *PAR* for each grid cell during each of these four stages within the 12-month period, referred to as $VI_{Ano}$ and $PAR_{Ano}$ respectively. The anomaly represents the departure from the average of the same month(s) during the non-drought years. These values were standardized by division over the corresponding standard deviation for non-drought years. We overlaid the spatial maps of $VI_{Ano}$ and $PAR_{Ano}$ to determine the percentage of grid cells where both anomalies moved in the same direction. Without drought stress, greenness should be controlled by solar radiation (Nemani et al., 2003; Huete et al., 2006; Saleska et al., 2016), with a positive radiation anomaly leading to a positive greenness anomaly. Then we considered the regions where the normal hydrological and thermal ranges were exceeded by overlaying their spatial distribution with that of $VI_{Ano}$ and $PAR_{Ano}$. Within these regions exceeding the 'normal' ranges, we counted the percentage of grid cells with below-average greenness; outside these regions, we counted the grid cells where $VI_{Ano}$ and $PAR_{Ano}$ had the same direction. The sum of these two percentage values was compared with that of only considering $VI_{Ano}$ and $PAR_{Ano}$.

## 4 Results

We found strong spatial and seasonal variations in the *TWS*, *LST* and *VPD* for non-drought years (Fig. 4). The values for September, December, March and June (Fig. 4a-l), illustrate the 12-month seasonal cycle. The minimum $TWS_{ND-Ave}$ (i.e. $TWS_{ND-Min}$) was observed around September in the south of the Amazon, and between December-March in the north (Fig. 4m-o). The maximum $LST_{ND-Ave}$ ($LST_{ND-Max}$) was observed around September for nearly all grid cells. Maximum *VPD* values ($VPD_{ND-Max}$) occurred around September in the southeast of the Amazon, and between December-March for part of the northwest.

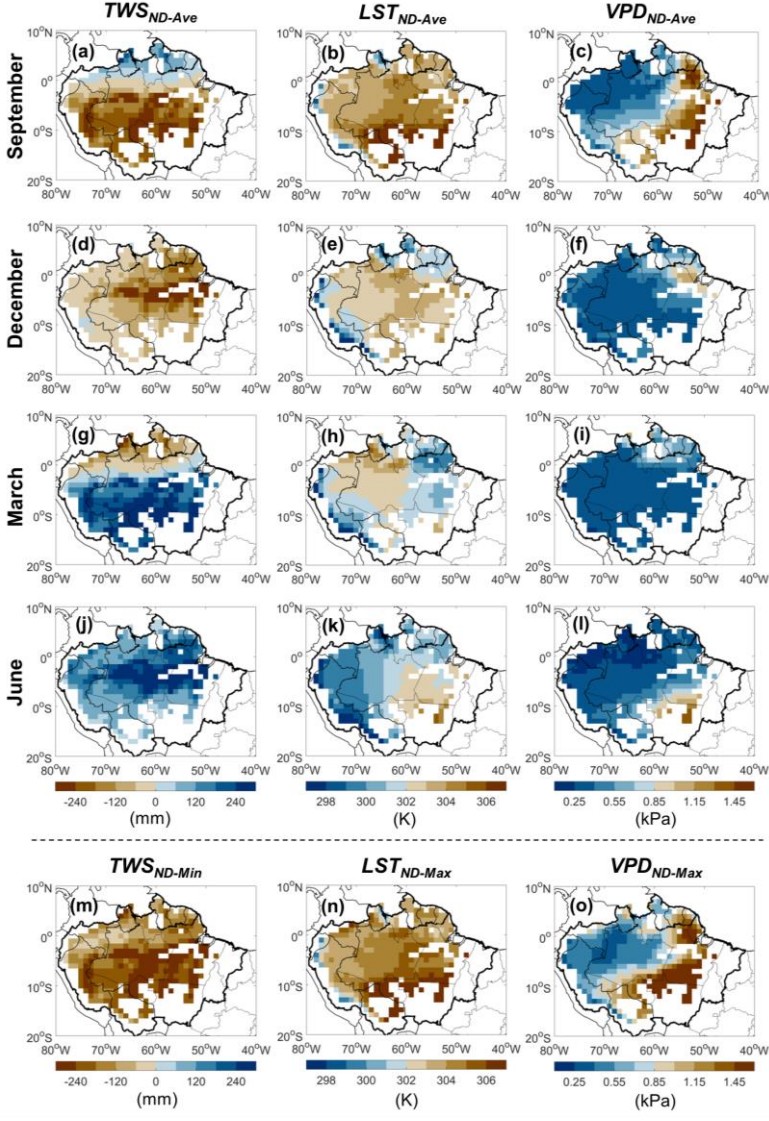

**Figure 3.** Spatial distribution of monthly average of non-drought years (ND) and extreme value of non-drought years' average over the 1° grid cells with more than 80% covered by 'evergreen broadleaf forests'. Panels (a) to (l) provide the spatial distribution of the average values of non-drought years, i.e. $TWS_{ND\text{-}Ave}$, $LST_{ND\text{-}Ave}$ and $VPD_{ND\text{-}Ave}$, for September, December, March and June, respectively. Panels (m), (n) and (o) show the spatial distribution of $TWS_{ND\text{-}Min}$, $LST_{ND\text{-}Max}$, and $VPD_{ND\text{-}Max}$, respectively.

The greatest departures of monthly *TWS*, *LST* and *VPD* during the 2015/16 drought occurred in different months (Fig. 4). *TWS* declined throughout the first half of the drought (Fig. 4a). Regional mean *TWS* was slightly above non-drought years' average during the first three months due to carryover of stored water from the wet preceding months (Fig. B1). *TWS* reached its lowest value in December 2015 and started to increase afterwards. Regional mean *LST* and *VPD* showed similar temporal dynamics (Fig. 4b-c). Both were higher than the non-drought years' average values throughout the full 12 months. The greatest *LST* and *VPD* anomaly departures occurred during Stage I (August–October 2015) and exceeded the 'normal'





range. They subsequently declined to within 'normal' range during Stage II (November 2015–February 2016) and moved closer to average values during Stage III (March–June 2016), before increasing again during Stage IV (July 2016). Summarizing, Stage I was characterized by high *LST* and *VPD* values above 'normal' ranges (Fig. 4d), while Stage II saw all three variables outside 'normal' ranges. Few grid cells with strong anomalies were detected during Stage III, but the number
of grid cells with *LST* and *VPD* outside 'normal' ranges increased again during Stage IV.

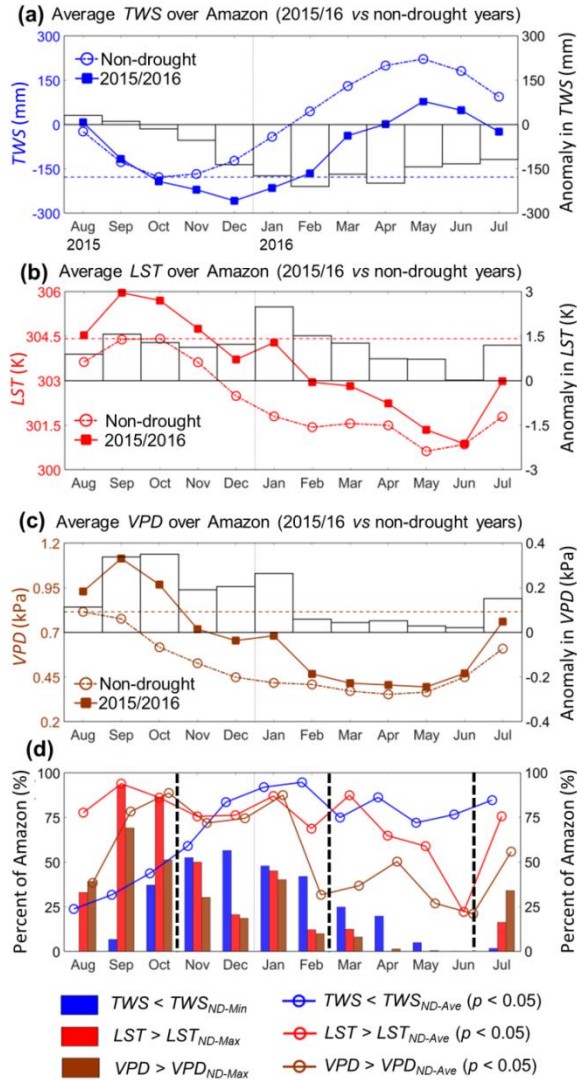

**Figure 4.** Temporal patterns of terrestrial water storage (*TWS*), land surface temperature (*LST*) and vapor pressure deficit (*VPD*) anomalies during the 2015/16 drought event. Panel (a) shows the regional average (i.e. average over all grid cells) *TWS* for each month from August 2015 to July 2016 as well as for the non-drought years' average (plot on left y-axis) and differences between the *TWS* values in 2015/16
and the non-drought years' average (bar on right y-axis). Panels (b) and (c) are the same as (a), but for *LST* and *VPD*, respectively. Panel (d) illustrates the percentage of Amazon rainforest exceeding the average and extreme values of non-drought years (ND). The colored bars show the percentage of Amazon rainforest with $TWS < TWS_{ND-Min}$, $LST > LST_{ND-Max}$ and $VPD > VPD_{ND-Max}$. The colored time series plots show the percentage of Amazon rainforest with statistically significant ($p < 0.05$) anomalies in *TWS*, *LST* and *VPD*.





Grid cells and drought stages were identified where *TWS, LST* and *VPD* (1) exceeded the extreme values of non-drought
years' 'normal' range, or (2) significantly ($p < 0.05$) departed from the average of non-drought years (Fig. 5). During Stage I,
*LST* exceeded $LST_{ND-Max}$ across the region while *VPD* exceeded $VPD_{ND-Max}$ over the central and north-east of the Amazon.
Stage II showed strong anomalies in *TWS, LST* and *VPD* and all three exceeded the 'normal' range in the north-central
region. During Stage III, only a small area with $TWS < TWS_{ND-Min}$ occurred in the north-east. During Stage IV, *LST* and *VPD*
exceeded the 'normal' range in the south of the Amazon. Thus, there was a gradual southwards movement of the regions
exceeding the 'normal' range, from the northeast during August–October 2015, to the central-north during November 2015–
February 2016, and finally the south by July 2016.

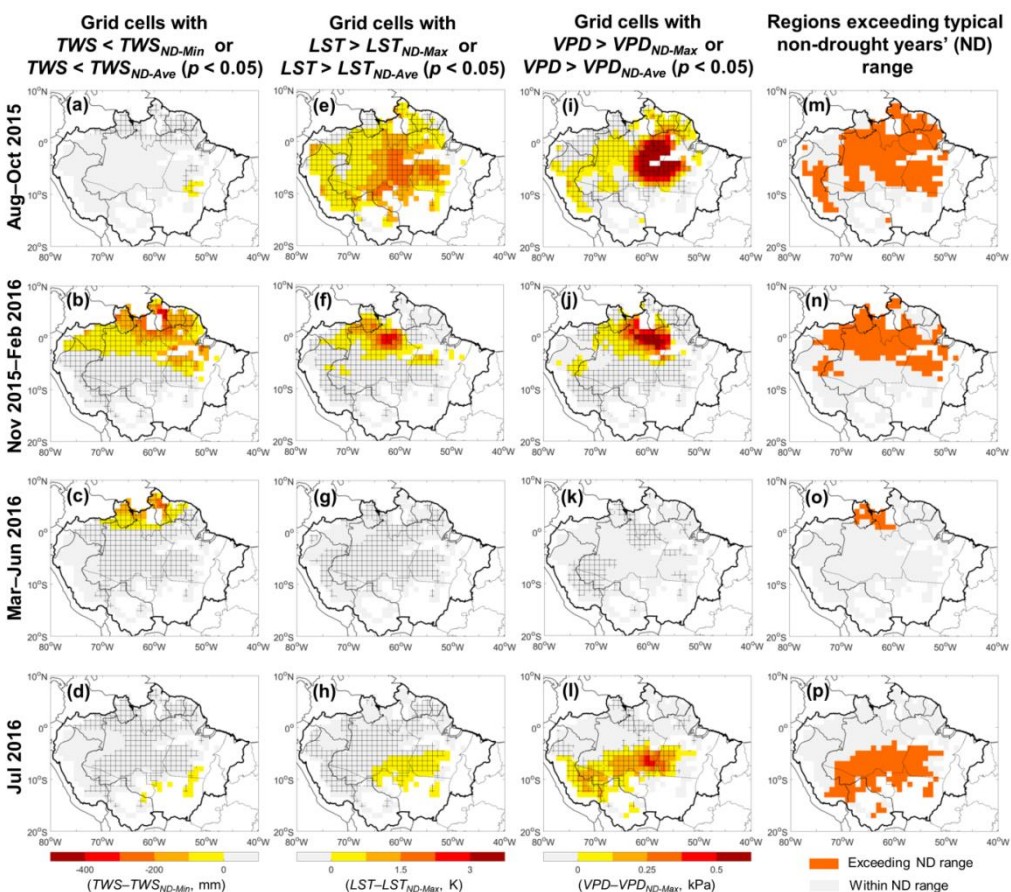

**Figure 5.** Spatial distribution of terrestrial water storage (*TWS*), land surface temperature (*LST*) and vapor pressure deficit (*VPD*) relative
to the non-drought years' (ND) average values ($TWS_{ND-Ave}$, $LST_{ND-Ave}$ and $VPD_{ND-Ave}$) and extreme values ($TWS_{ND-Min}$, $LST_{ND-Max}$ and
$VPD_{ND-Max}$), over the 1° grid cells with more than 80% covered by 'evergreen broadleaf forests'. The first column from the left shows the
overlay between the grid cells where $TWS < TWS_{ND-Min}$ (in red-to-yellow color) and the grid cells with *TWS* statistically significant ($p <$
0.05) below $TWS_{ND-Ave}$ (hatched area) over the period of (a) August–October 2015, (b) November 2015–February 2016, (c) March–June
2016, and (d) July 2016. The second and third columns show the same, but for *LST* and *VPD*, respectively. The last column shows the
regions where the normal hydrological and thermal ranges of non-drought years were exceeded for four stages, respectively, according to
the criteria defined in the Methods section.



We compared the spatial distribution of grid cells where the normal hydrological and thermal ranges were exceeded (Fig. 5) to that of *VI* and *PAR* anomalies for each of the four drought stages (Fig. 6a-d). It appeared drought and radiation can explain 70% of observed greenness anomalies (Table 2). This is an improvement over considering only the anomalies in *VI* and *PAR*

(right column in Fig. 6), with increase by 33% in Stage I and 28% in Stage IV (Table 2). Moreover, for all grid cells with variables exceeding the 'normal' ranges during these four stages, 75% coincided with below-average greenness.

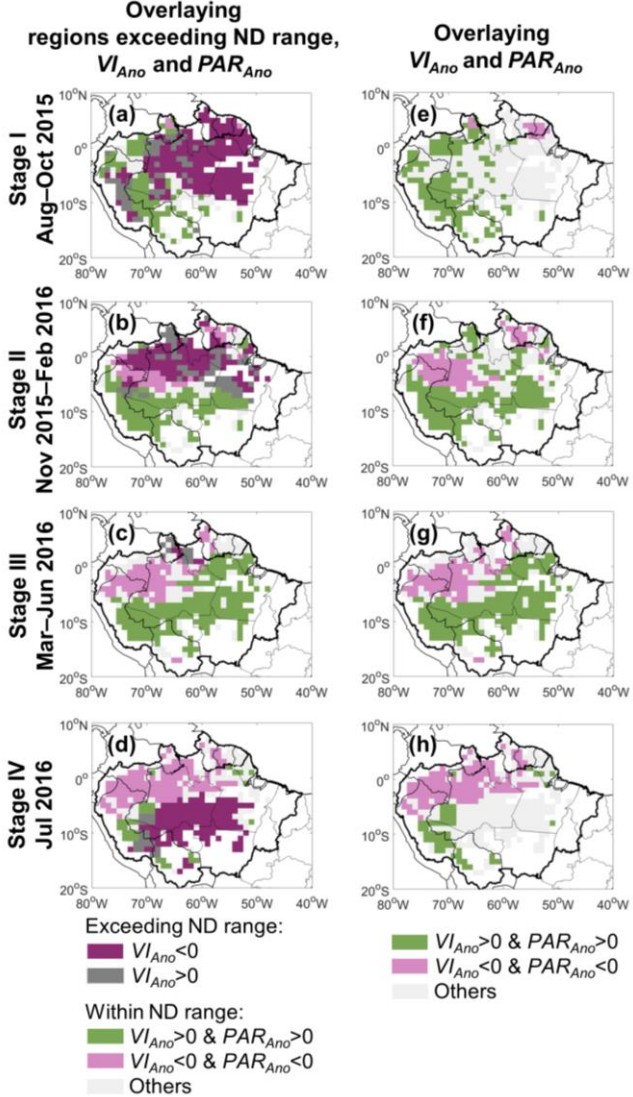

**Figure 6.** Spatial agreements of regions exceeding the normal hydrological and thermal ranges of non-drought years (ND), vegetation indices anomaly ($VI_{Ano}$) and photosynthetically active radiation anomaly ($PAR_{Ano}$) for the four stages of the August 2015–July 2016

drought, over the 1° grid cells with more than 80% covered by 'evergreen broadleaf forests'. Panels (a) to (d) in the left column include the regions exceeding non-drought years' range on top of $VI_{Ano}$ and $PAR_{Ano}$. Within these regions, the grid cells with negative $VI_{Ano}$ were counted (dark purple); outside of these regions, the grid cells where $VI_{Ano}$ and $PAR_{Ano}$ had the same anomaly direction were counted (green and light purple). Panels (e) to (h) in the right column overlay $VI_{Ano}$ and $PAR_{Ano}$ for each of four stages. The grid cells where $VI_{Ano}$ and $PAR_{Ano}$ had the same anomaly direction were counted (green and light purple).



**Table 2.** Percent of Amazon (%) where greenness anomaly associated with different factors

| Period | (i) Regions exceeding non-drought years (ND) range, $VI_{Ano}$ and $PAR_{Ano}$ | (ii) $VI_{Ano}$ and $PAR_{Ano}$ | Difference between (i) and (ii) |
|---|---|---|---|
| Stage I (August–October 2015) | 72% | 39% | +33% |
| Stage II (November 2015–February 2016) | 68% | 66% | +2% |
| Stage III (March–June 2016) | 72% | 72% | 0% |
| Stage IV (July 2016) | 71% | 44% | +28% |

## 5 Discussion

The spatiotemporal patterns of canopy greenness anomaly during the 2015/2016 drought found herein agree well with other independent satellite- and field-based vegetation observations. From the perspective of satellite observations, Koren et al. (2018) used the newly developed satellite-based sun-induced fluorescence (SIF) product (2007–2016) to examine the impact
of the 2015/2016 Amazon drought. Temporally, it was found that the regional mean SIF was below its climatological average at the beginning and end of the drought, but above the average in the first half of 2016. Spatially, the eastern part of Amazon experienced much larger reductions in SIF than the western part. Petchiappan et al. (2022) used the Advanced Scatterometer (ASCAT) backscatter (2007–2016) and found large-scale negative anomalies in backscatter over the Amazon rainforest and savannah in late 2015, with a stronger magnitude over the eastern part of the region. From the perspective of
field measurements, Santos et al. (2018) measured leaf gas exchange, chlorophyll and nutrient content in canopy leaves in the central Amazon throughout 2015 and during the dry season of 2016. They found that, during the extremely dry season of 2015 under conditions of extremely high *LST* and *VPD*, the light-saturated photosynthetic rate decreased 28%, relative to other 2015 seasons and the dry season of 2016. However, with precipitation returning after the dry season of 2015, the photosynthetic rate increased to 'normal' conditions again. Meanwhile, massively new leaf flushing occurred, leading to
above-average canopy greenness in the first half of 2016 (Goncalves et al., 2020). As for the possible causes for the quick recovery of photosynthetic rate, Santos et al. (2018) found that the photosynthesis reduction under extreme drought and high temperature in the 2015 dry season was primarily due to stomatal closure, which can reverse when water becomes available.

Our *TWS*, *LST* and *VPD* based threshold approach developed herein is also supported by findings from field measurements
during the 2015/16 Amazon drought. Fontes et al. (2018) found that leaf and xylem safety margins (LXSMs) of central Amazonian trees showed a sharp drop in the months with unusually high canopy temperature and *VPD* during the 2015/16 drought. LXSMs were significantly negatively ($p < 0.05$) correlated with *VPD*, but not with soil water storage. Moreover, the




high values of predawn leaf water potential from 2015 through 2017 suggested that soil water supply was not limiting during their study period. These results indicate that the atmospheric demand could be the main driver for plants' LXSMs decreases. Meng et al. (2022) showed that the rapid decline in sap velocity during the 2015/16 Amazon drought was accompanied by a marked decline in soil moisture and an increase in air temperature and *VPD*, and provided evidence for a soil water threshold below which sap velocity shifted from light-limited to water-limited. Therefore, the combination of *TWS, LST* and *VPD* provided the means for the regions and periods dominated by high atmospheric demand or/and low soil water availability to be identified.

The spatiotemporal analysis approach developed here shows both similarities and differences with the Maximum Climatological Water Deficit (MCWD) approach commonly used to characterize water stress during droughts at large scale across Amazon rainforest (Aragão et al., 2007; Lewis et al., 2011; Aragão et al., 2018). An important difference is that MCWD is calculated using a simple bucket model approach, with a running water balance from monthly precipitation and an assumed constant actual evapotranspiration of 100 mm per month (da Rocha et al., 2004; Guan et al., 2015; Maeda et al., 2017). It makes no assumption of soil water storage in calculating a water 'deficit'. When monthly precipitation is below 100 mm, the calculated water deficit of that month is the difference between precipitation and evapotranspiration (negative value). When monthly precipitation is above 100 mm, water deficit of that month is calculated as the difference between precipitation and evapotranspiration (positive value) plus the water deficit of the previous month; if this sum-up is above zero, it is set to zero. Accordingly, calculated in this way without any soil water storage term (Meir et al. 2015), the water deficit can become a very strongly negative value when precipitation is below 100 mm for several months in a row. The MCWD corresponds to the maximum value of the water deficit reached for a grid cell within the year. The MCWD anomaly, i.e. the difference in MCWD between drought and non-drought years, is used to characterize the severity of water stress. The MCWD approach is therefore a measure of deficit in the water 'flux' during the drought year, i.e. how much less water falls into the soil consecutively over time, whereas the method we present here focuses on the water storage 'status' at monthly to seasonal time scales, i.e. when and where the water storage is below the minimum level of non-drought years. These two approaches provide complementary information. To illustrate the differences that arise from the two approaches, we calculated the MCWD anomaly over the Amazon for the 2015/16 drought year following Aragão et al. (2007) (Fig. C1). The strongest calculated MCWD anomaly occurred over the north-central Amazon, which agrees with the location of anomalies in our observation-based water availability data ($TWS < TWS_{ND\text{-}Min}$) during Stages II and III (Fig. 5). Considering both fully independent information sources together provides corroborating evidence and supports a more robust characterization of water availability during drought. A further difference is that we also took *LST* and *VPD* conditions into account. We identified regions where high *LST* and *VPD*, rather than a water deficit per se, appeared to be the main drivers associated with below-average canopy greenness during Stages I and IV (Fig. 5).





Our results demonstrate that comparing values of *TWS*, *LST* and *VPD* to their non-drought years' ranges can help delineate the most likely drought-affected regions and explain spatiotemporal patterns in greenness anomalies. There are a number of caveats to the method and data used, and these may be responsible for some of the remaining 30% of unexplained greenness anomalies. Firstly, each of the datasets used has its uncertainties. These certainly include uncertainties in vegetation indices
due to sun-target-sensor geometry and atmospheric effects, but also uncertainties in the other data used. Secondly, we used the range of *TWS*, *LST* and *VPD* in non-drought years as an estimate of the tolerance thresholds of the rainforest. This is a simplified representation, as a sharp threshold is not to be expected given the ecological and physiographic complexity of the large areas covered by each grid cell. It is also possible that the observed non-drought years' ranges of variables were exceeded without in fact exceeding physiological and ecological tolerance thresholds in the vegetation. In that case, for
example, higher *VPD* would act to enhance rather than limit photosynthesis and lead to above- rather than below-average greenness. Thirdly, there may be additional local factors controlling greenness that are not captured in the satellite and re-analysis data record. Finally, the non-drought years' range defined here is based on a relatively short record in relation to the effect the lifespan of the dominant rainforest vegetation and how natural selection may act to alter the related ecological thresholds, and so this 'normal' range should be considered a qualitative estimate. With the availability of longer and more
reliable satellite records, along with increasing ground-based observations, it should become possible to develop a more sophisticated approach to quantify, predict and interpret the response of the Amazon rainforest to combined water, heat and radiation conditions during future droughts.

## 6 Conclusions

We developed a 'normal' range-based approach to delineate the regions where the normal environmental ranges experienced
during non-drought years were exceeded during the 2015/16 year-long Amazon drought, focusing on three main environmental metrics: terrestrial water storage, land surface temperature and atmospheric moisture demand records covering 2003–2016. We found a gradual southwards shift of these regions: from (1) the north-eastern Amazon during August–October 2015 mainly due to high temperatures and high atmospheric moisture demand; to (2) the north-central during November 2015–February 2016 where soil water deficit, high temperatures and high atmospheric moisture demand
co-existed simultaneously; and (3) the southern in July 2016 caused by high temperatures and high atmospheric moisture demand again. Within these regions, 75% of all grid cells were characterized by below-average greenness determined from MODIS vegetation index. Outside of these regions, greenness anomalies and radiation anomalies were generally in phase, which is expected to occur under normal conditions. Combined, drought impact and radiation anomalies can explain more than 70% of the observed swing pattern in the regional greenness, i.e., below-average values during the onset and end of the
drought but above-average values during the intervening months. These results suggest that our method of combining water storage, temperature and atmospheric moisture demand together can reasonably identify the most likely drought-affected regions at monthly to seasonal time scales during an event such as the 2015/16 El Niño. Our analysis also highlights the





necessity to take into account whether the long-term normal hydrological and thermal ranges were exceeded when interpreting the response of Amazon rainforest to droughts in the future.

**Appendix A Gap-filling of TWS**

We gap-filled the missing values in the original terrestrial water storage (*TWS*) dataset over the Amazonia individually for each 1° spatial resolution grid-cell. A time series of monthly precipitation (*P*), photosynthetically active radiation (*PAR*) and original terrestrial water storage (*TWS*) from January 2003 through December 2016 for an example grid-cell from southern Amazonia is shown in Fig. A1. There are 168 months in total for this 14-year period and *TWS* values are missing for 21

months. The gap-filling of missing *TWS* values is based on the principle that the change in *TWS* (i.e., time step *t* minus time step *t-1*) is highly related with *P* and *PAR* at the time step *t*. Here a multiple linear regression equation is used to establish the relationship of these variables for each grid-cell.

$$\textit{Change in TWS (t) = TWS(t) – TWS(t-1) = a x P(t) + b x PAR(t) + c} \quad \text{(A1)}$$

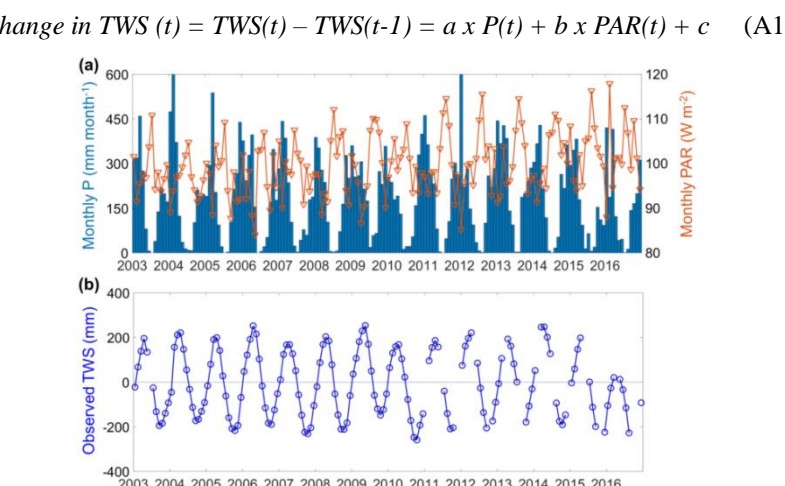

**Figure A1.** Example illustrating the monthly time series of (a) precipitation (*P*), photosynthetically active radiation (*PAR*) and (b) original terrestrial water storage (*TWS*) from January 2003 through December 2016 for the grid-cell centered at 7.5°S and 55.5°W. Over this 168-month period, TW*S* are missing for 21 months (the longest gap is 3 months) while no *P* or *PAR* are missing.

There are 131 valid values of "change in *TWS*" for the example grid-cell (i.e., N=131). By fitting the multiple linear equation, the values for parameter a, b and c are 0.47, -0.48 and -41.5, respectively, with the resulting correlation coefficient

(*R*) of 0.89 and root mean square error (*RMSE*) of 34.8 mm (Fig. A2a). After moving the term *TWS(t-1)* to the right of the equation, we can compare the observed *TWS* (i.e., *TWS(t)*) with the estimated *TWS* based on *P(t)*, *PAR(t)* and *TWS(t-1)* (see Fig. A2b). The *R* and *RMSE* values between them are 0.98 and 32.5 mm, respectively. The missing *TWS* values at time step *t* can then be estimated according to the equation 0.47x*P(t)*-0.48x*PAR(t)*-41.5+*TWS(t-1)*, and the gap-filled *TWS* time series is shown in Fig. A2c. Our approach is able to estimate and gap-fill the maximum and minimum monthly value of a year (e.g.,

in 2013, 2015 and 2016), which is difficult for linear interpolation approach.




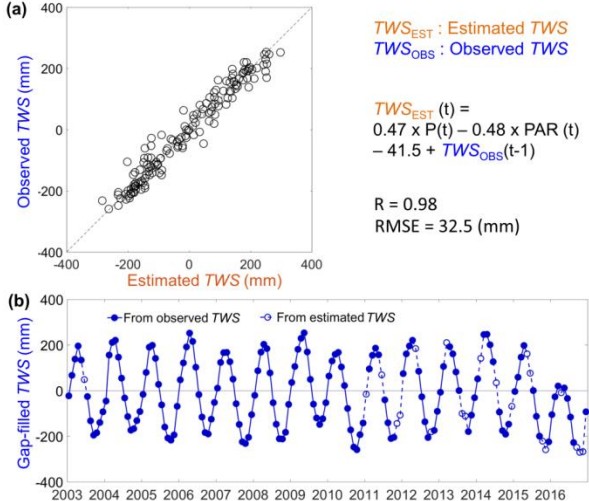

**Figure A2.** (a) Scatterplot of (y-axis) observed *TWS* and (x-axis) estimated *TWS* according to *P*, *PAR* and observed *TWS* from the previous time step. (b) Time series of gap-filled *TWS* by combining observed *TWS* and estimated *TWS*.

When we applied this gap-filling approach to each grid-cell over the Amazon rainforest independently, the estimated *TWS*
that we obtained are highly correlated with observed *TWS*, with *R* values higher than 0.90 over 90% and higher than 0.8 over
99% of the Amazon region (Fig. A3a). For the *RMSE* between observed and estimated *TWS*, one third of the Amazonia has
the value below 40 mm, and two thirds are lower than 50 mm (Fig. A3b). Higher *RSME* values are found along the major
rivers where the dynamic ranges of *TWS* are also higher (Fig. A3b and c). Overall, the estimated *TWS* for the missing time
steps, based on *P*, *PAR* and observed *TWS* from the previous time step, are reasonable.

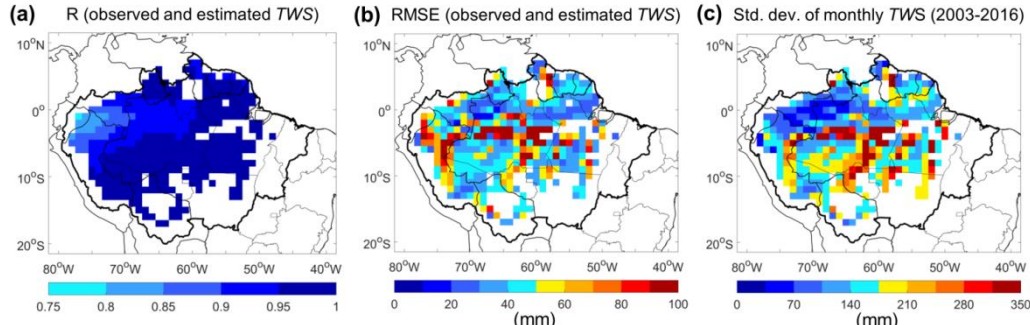


**Figure A3.** Spatial distribution of (a) *R* and (b) *RMSE* between observed *TWS* and estimated *TWS*, as shown in Fig. A2a, and (c) standard deviation value of monthly *TWS* from 2003 to 2016, over 1° grid cells having more than 80% of 0.05° IGBP grid cells classified as 'evergreen broadleaf forests'.



**Appendix B TWS anomaly immediately preceding the 2015/16 drought**

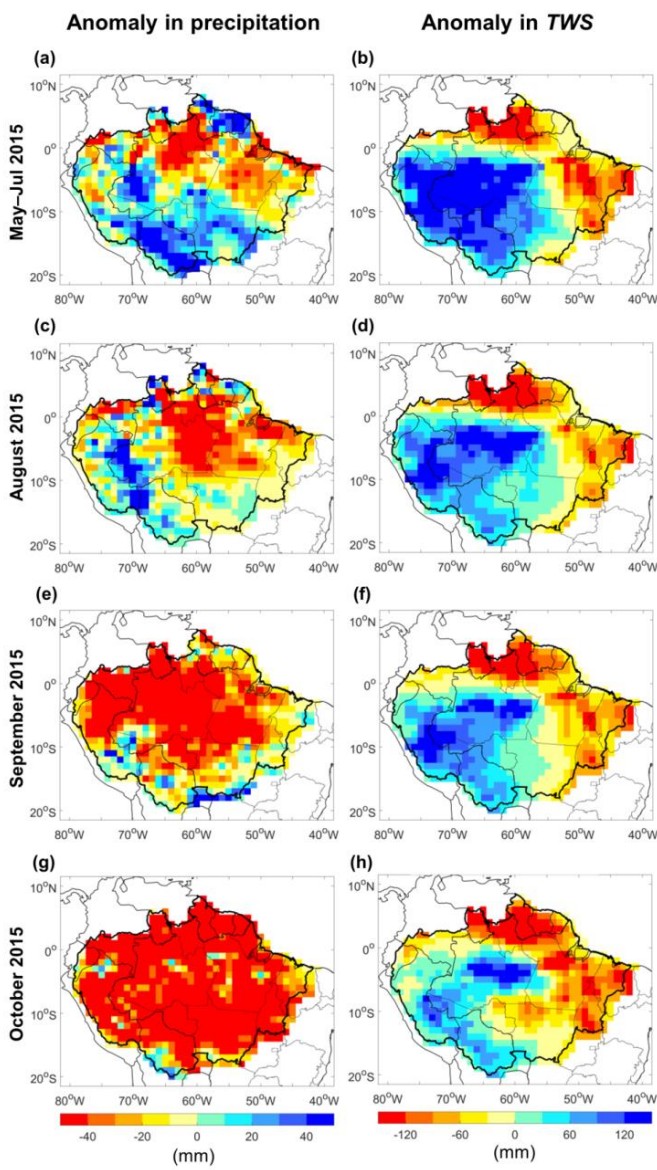

**Figure B1.** Spatial distribution of anomaly in precipitation and *TWS* during (a and b) May–July 2015, (c and d) August 2015, (e and f) September 2015, and (g and h) October 2015, respectively. It can be seen that although precipitation was below average during August–October 2015, above-average *TWS* was still observed over western part of Amazon, due to the carryover effect of above-average *TWS*
from May-July 2015.



## Appendix C MCWD anomaly during August 2015–July 2016

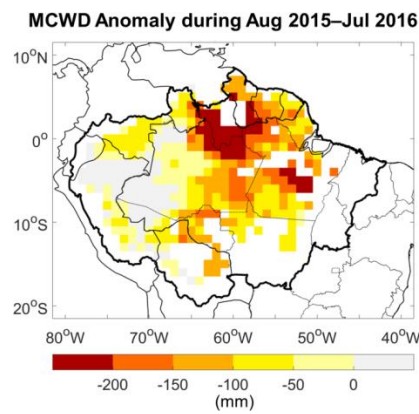

**Figure C1.** The difference between MCWD during August 2015–July 2016 and the mean MCWD of non-drought years (2003–2016, excluding 2005, 2010, 2015 and 2016) over the 1° grid cells with more than 80% covered by 'evergreen broadleaf forests'. MCWD stands for maximum climatological water deficit, and its calculation can be found in Aragão et al. (2007). The monthly precipitation data used here is derived from TRMM (TRMM 3B43 v7, see Table 1).

## Data availability

All data used in this paper are present in Table 1 with download links provided. Additional information associated with the paper is available from the corresponding author upon request.

## Author contribution

All authors conceptualized the study. YYL conducted the analysis and wrote the first draft of the manuscript, with subsequent addition and improvement by all authors.

## Competing interests

The authors declare that they have no conflict of interest.

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
