# Peer review of "Drought and radiation explain fluctuations in Amazon rainforest greenness during the 2015–2016 drought"

_Biogeosciences, 2023_

## Author Response (AR1)

Dear Prof. Rammig / Prof. Vergopolan,

First, we would like to thank you, Dr Baker and the other anonymous referee for the constructive comments and suggestions on BG-2023-155, which were very helpful for us to improve the quality of our manuscript. All reviewer comments have been positively implemented in the revised manuscript. Below are our responses to the referees' comments. The comments are in black, and our responses are in blue. In our response letter, codes are used for each comment to assist with navigation; for example, R1C2 means 'Reviewer #1 Comment #2' and so on for all other comments.

We hope you and the referees are satisfied with our revisions, and if you have any additional questions or comments, please feel free to contact me. We look forward to hearing from you soon.

Best regards,

Yi LIU on behalf of all co-authors

**Referee #1**

The manuscript of Liu et al. is an interesting study about identifying the different environmental drivers of the drought-affected regions of 2015-2016 within the Amazon forest. They show that the regions where water storage, temperature and atmospheric moisture demand exceeded their 'normal' ranges agreed with more than 70% of the observed patterns in vegetation greenness. This manuscript has potential and could be a useful contribution to the drought research in the tropics. I do have some questions that I would like to see addressed.

**General comments**

**R1C1:** I am not convinced by how the authors defined the occurrence of extreme values of LST, VPD and TWS in 2015-2016. Extreme values of 'normal' years are defined as the lowest or highest mean monthly value, not as the actual lowest or highest value that a 'normal' month reached for a certain variable. If there is a lot of variation in monthly TWS, LST or VPD between different years, the mean value doesn't catch that. Comparing a raw value to a mean without taking into account the normal variation will easily give 'extreme' results. The authors combine this metric of extremeness with a significant difference from the monthly mean using a Wilcoxon rank test. I am curious how different the results would be when only one of the requirements is used (i.e. only using the requirement of at least two variables that are more extreme than the normal extreme, or of one extreme variable and the other two being significantly different)? Do they both give similar results or is one more in agreement with the vegetation anomalies than the other? This analysis could be added in the appendix.

**Response:** Following suggestions from both referees, we conducted a thorough revision, including a sensitivity analysis which we now include in the main text of the revised manuscript. To better explain what we did, we combine our responses to R1C1 and R1C2 here, which consists of three parts.

Part I

In the revised manuscript, an explanatory figure (please see our new Figure 2) depicting this analysis design was added to the Methods section for enhanced clarity.

[Figure]

**Figure 2.** Overview of the experimental design implemented herein. Examples with 9 grid cells are used here to illustrate how the directions of VI anomalies were predicted for each grid cell in these two approaches. Developing the terrestrial water storage (TWS), land surface temperature (LST) and vapor pressure deficit (VPD)-based method to categorize grid cells into two groups in Approach #2 is the focus of the Methods section. The impact of precipitation variability (e.g. total annual precipitation, length of dry season) is accounted for by these three variables, and therefore, precipitation is not included in the method in Approach #2.

Below are the new paragraphs associated with this new figure in the revised manuscript.

[revised manuscript text omitted]

Part III

In the Results section, we compared the predicted VI anomaly direction (derived from Approaches #1, #2A, #2B and #2C) and MODIS-observed VI anomaly direction and quantified their performance.

Below are the new figure, table and associated paragraphs.

'Spatial distributions of predicted VI anomaly direction (derived from Approaches #1, #2A, #2B and #2C) and MODIS-observed VI anomaly direction for the four stages from August 2015 through July 2016 are shown in Fig. 8. Their spatial agreements (%) are shown in Table 2. When compared with approach #1, all three #2 approaches have a better spatial agreement with MODIS observations, with the best performance derived from #2C.'

[Figure]

Predicted *vs* Observed VI anomaly direction

**Figure 8.** Spatial distributions of (1ˢᵗ-4ᵗʰ columns) predicted VI anomaly direction from approaches #1, #2A, #2B and #2C, respectively, and (5ᵗʰ column) MODIS-observed VI anomaly direction for the four stages from August 2015 through July 2016.

**Table 2.** Spatial agreement (%) between predicted VI anomaly direction derived from different approaches and MODIS-observed VI anomaly direction. There are 390 one-degree grid cells over the Amazon with more than 80% covered by 'evergreen broadleaf forests' considered in these statistics.

| Period | Approach #1 (Using PAR) | Approach #2A | Approach #2B | Approach #2C |
| --- | --- | --- | --- | --- |
| | | (Using TWS, LST and VPD first, then PAR) | | |
| Stage I (August–October 2015) | 39% | 67% | 54% | **72%** |
| Stage II (November 2015–February 2016) | 66% | 68% | 68% | **68%** |
| Stage III (March–June 2016) | 72% | 72% | 72% | **72%** |
| Stage IV (July 2016) | 44% | 59% | 69% | **71%** |

**R1C2:** It is not clearly explained in the methodology or the results how the regions with VI and PAR anomalies are compared with the extreme VPD, TWS and LST regions. This is the main analysis that leads to the conclusions of the manuscript, so it is important that this part is clear.

**Response:** A new explanatory figure (new Figure 2) was added to the Methods section in the revised manuscript to explain how we compared the PAR anomaly-based VI anomalies and TWS, LST and VPD-based VI anomalies. Please see Part I of our response to R1C1 above. Thanks very much for your comments, and we hope our methodology is now clearly explained.

**Specific comments**

**R1C3:** I would change the word 'swings' in the title. It is not a very common scientific word and it does not get repeated once within the manuscript, making it a strange wording choice.

**Response:** Done. We replaced the word 'swings' with 'fluctuations'. Accordingly, the title becomes 'Drought and radiation explain fluctuations in Amazon rainforest greenness during the 2015–2016 drought'.

**R1C4:** The abstract feels too technical and methodological. I would make the explanation of the approach shorter, so that the conclusion of the manuscript is more clear. As a reader, my attention would get lost in the technicalities of the abstract and it would not convince me to read the paper (which would be a pity, because the research is valuable).

**Response:** Done. Following the referee's comment, we replaced the technical parts from line 21 to 26 by one sentence 'we proposed an approach to categorize regions into two groups: (1) those exceeding normal hydrological and thermal ranges and (2) those within normal ranges'.

Below is the revised abstract.

'The 2015/16 Amazon drought was characterized by below-average regional precipitation for an entire year, which distinguishes it from the dry-season-only droughts in 2005 and 2010. Studies of vegetation indices (VI) derived from optical remote sensing over the Amazonian forests indicated three stages in canopy response during the 2015/16 drought, with negative VI anomalies during the onset and end of the drought, and positive VI anomalies during the intervening months. To date, a satisfactory explanation for this broad temporal pattern has not been found. A better understanding of rainforest behaviors during this unusually long drought should help predict their response to future droughts. We hypothesized that negative VI anomalies could be caused by water and heat stress exceeding the tolerance ranges of the rainforest. To test our hypothesis, based on monthly observations of terrestrial water storage (TWS), land surface temperature (LST) and vapor pressure deficit (VPD) for January 2003–December 2016, we proposed an approach to categorize regions into two groups: (1) those exceeding normal hydrological and thermal ranges; and (2) those within normal ranges. Accordingly, regions exceeding normal ranges during different stages of the 2015/16 event were delineated. The results showed a gradual southward shift of these regions: from the north-eastern Amazon in August–October 2015, to the north-central part in November 2015–February 2016 and finally to the southern Amazon in July 2016. Over these regions exceeding normal ranges during droughts, negative VI anomalies were expected, irrespective of radiation anomalies. Over the regions within normal ranges, VI anomalies were assumed to respond positively to radiation anomalies, as is expected under normal conditions. We found that our proposed approach can explain more than 70% of the observed spatiotemporal patterns in VI anomalies during the 2015-16 drought. These results suggest that our 'exceeding normal ranges'-based approach combining (i) water storage, (ii) temperature, and (iii) atmospheric moisture demand drivers can reasonably identify the most likely drought-affected regions at monthly to seasonal time scales. Using observation-based hydrological and thermal condition thresholds can help with interpreting the response of the Amazon rainforest to future drought events.'

**R1C5:** The authors use a combination of NDVI and EVI to quantify the greenness anomalies of the forest, but there is no mention of the problem with NDVI saturation in high biomass regions.

**Response:** Done. in the revised manuscript, we added one figure (Fig. D1) and associated text to address this point.

'The NDVI may exhibit the signal saturation issue over high biomass regions (Huete et al. 2002). We examined the anomaly in NDVI and EVI separately and found their spatial distributions are similar (Fig. D1). Therefore, we combined NDVI and EVI to quantify the greenness anomalies in this study.'

**Figure D1.** Standardized anomalies in (a) EVI and (b) NDVI during the 2015/16 Amazon drought over the 1° grid cells with more than 80% covered by 'evergreen broadleaf forests'. EVI and NDVI anomalies show the same anomaly direction over 70% of these grid cells.

**R1C6:** Line 191-193: Why did you use the non-parametric Wilcoxon rank test instead of calculating standardized anomalies and using them to say when values were significantly different?

Response: Done. We added the sentence below to the revised manuscript.

'As many hydrologic variables are not normally distributed, using the non-parametric Wilcoxon rank test offers the advantage of not assuming that data are normally distributed.'

**R1C7:** Line 195-198: How were these requirements decided? Based on literature (if so, add references)? Based on which requirements gave the highest agreement with the vegetation indices (if so, add the other methods that were tried)?

**Response:** Done. During the revision, we tested three ways to determine the hydrological and thermal conditions of a grid cell 'exceeding normal ranges'. Accordingly, we added one figure in the revised manuscript (new Figure 4 below) to illustrate our tested three ways. Please see Part II of our response to R1C1 above.

**R1C8:** Line 208-212: Step 3 in methods is not very clear. It might be better to add an explanatory figure such as figure 2? Too much mixed vocabulary used: 'positive greenness anomaly', 'below-average greenness', 'exceeding the normal range' à this makes it difficult to follow. Maybe better to use one word for 'regions exceeding the normal ranges', to make the explanation less wordy.

**Response:** Done. We thoroughly revised the Methods section during the revision. We added an explanatory figure (new Figure 2 below) depicting this analysis design for enhanced clarity. Please see Part I of our response to R1C1 above.

**R1C9:** Line 211-212 is not clear: 'compared with that of only considering VI-ano and PAR-ano'. What does this mean?

**Response:** Done. In the revised manuscript, we added an explanatory figure (new Figure 2) depicting this experimental design. VI-ano and PAR-ano become Approach #1 in the new figure. Please see Part I of our response to R1C1 above.

**R1C10:** Figure 4 does not completely convince me that there are, for example, four months with an extremely low TWS in 2015-2016. How extreme is this compared to the normal variation in non-drought years? Figure 2b-d shows that there are indeed large variations in the monthly values. Maybe add the standard deviation in lines around the monthly dots? This will probably make the figure too crowded to be pretty, but this might be something for the appendix.

**Response:** Done. During the revision, we incorporated standard deviation lines around the non-drought years' monthly average dots to highlight the extreme nature of TWS, LST and VPD in 2015-2016. Please see the new Figure 6 below.

[Figure]

**Figure 6.** Temporal patterns of terrestrial water storage (TWS), land surface temperature (LST) and vapor pressure deficit (VPD) anomalies during the 2015/16 drought event. Panel (a) shows the regional average (i.e. average over all grid cells) TWS for each month from August 2015 to July 2016 as well as for the non-drought years' average (± standard deviation) (plot on left y-axis) and differences between TWS values in 2015/16 and non-drought years' average (bar on right y-axis). It is noted that we first calculated the regional average TWS for each month from January 2003 through December 2016, and then derived non-drought years' average and standard deviation. Panels (b) and (c) are the same as (a), but for LST and VPD, respectively.

**R1C11:** Table 2: it is not clear how the percentages are calculated. Is it 72% of all pixels with VI and PAR anomalies in same direction that also exceeded the ND range, or 72% of all pixels that exceeded the ND range that had VI and PAR anomalies in same direction, or …? This could be better explained in the table caption. I don't understand what the current table caption means.

**Response:** We re-designed Table 2 and all percentages in the new table represent the spatial agreement (%) between predicted VI anomaly direction derived from different approaches and MODIS-observed VI anomaly. Please see Part III of our response to R1C1 above.

**Technical corrections**

**R1C12:** Line 52: Add 'the' to the Amazon forest response.

**Response:** Done.

**R1C13:** Line 56: Add 'the' to the NDVI or remove 'the' from the EVI.

**Response:** Done.

**R1C14:** Line 101: Add 'the' to the 2005 drought.

**Response:** Done.

**R1C15:** Line 197: Remove 'being' from sentence.

**Response:** Done.

**R1C16:** Line 214-219: I think this part refers to Figure 3 instead of Figure 4?

**Response:** Done. Thanks for your careful review (and apologies for our error). We have corrected the figure number in the revised manuscript.

Thanks very much for your constructively critical comments that have been the catalyst for us to improve our manuscript. We appreciate the time and energy you've invested in reviewing our manuscript and have acknowledged your efforts in the appropriately named section.

**Referee #2**

**Summary:** The authors use a combination of TWS, VPD and LST (abbreviations explained in the manuscript, so I'm not going to rehash them here) to explore correlation between drought and greenness, as expressed by MODIS NDVI and EVI. They set a criteria of 2 of 3 variables beyond extreme values, or 1 variable beyond extreme and 2 significantly different from average as being outside of normal range. The idea (as I'm interpreting it) is that Amazonian forests have evolved to maintain ecophysiological function within certain bounds of water, temperature, and humidity stress, and the authors have defined a metric to determine when those bounds have been exceeded. They find that these metrics "explained more than 70% of the observed spatiotemporal patterns in greenness". When applied to the El Nino of 2015/2016 they find that drought as expressed in VIs in general moves from north to south during the event, and from August 2016 through July 2016 the regional VI progresses from below-normal, near-normal, above-normal, and below-normal stages.

**R2C1:** This paper is interesting and well-written, although a bit dense at times. It took me several readings to get my head around the method, but once I did I found it an interesting and thought-provoking paper. Initially I wondered if the authors were neglecting the precipitation variability in the region (total annual precipitation, length of dry season), but I realized that this variability is accounted for in the construction of the 'normal' cycles for each gridcell shown in Figure 2. The authors might spell this out explicitly-wouldn't take more than a couple of words or a sentence added. I do have a lot of questions about the methods and results, but I do not have any objections that would lead me to recommend rejection. As I don't believe that a major overhaul is required to address my comments, my formal recommendation is that this paper be accepted for publication with minor revisions.

**Response:** In the revised manuscript, an explanatory figure (new Figure 2) depicting this analysis design was added to the Methods section for enhanced clarity. Also, we added that the impact of precipitation variability (total annual precipitation, length of dry season) is accounted for by TWS, LST and VPD and, therefore, not included in the 'exceeding normal ranges' method.

[Figure]

**Figure 2.** Overview of the experimental design implemented herein. Examples with 9 grid cells are used here to illustrate how the directions of VI anomalies were predicted for each grid cell in these two approaches. Developing the terrestrial water storage (TWS), land surface temperature (LST) and vapor pressure deficit (VPD)-based method to categorize grid cells into two groups in Approach #2 is the focus of the Methods section. The impact of precipitation variability (e.g. total annual precipitation, length of dry season) is accounted for by these three variables, and therefore, precipitation is not included in the method in Approach #2.

**Review:** Let's get started…

**R2C2:** There has been an ongoing discussion (or debate, if you want) around the notion of greenness increasing with (mild) drought (Saleska, Huete) or not (Samanta, Morton) for the last 10 years or so. Saleska claims the argument over with the results of Wu, Albert, Restrepo-Coupe and others who find that there is (in parts of the Amazon) a drop and reflush of leaves at the end of the dry season, but the new leaves have reduced photosynthetic capacity until they have 'matured' for somewhere around 60-90 days. The authors mention this element of regional ecophysiology, but do so in a rather oblique manner. This paper would have much more impact if the issue was met head-on. The community knows about the debate and recent results, the subject matter of this paper is related to this topic, so why not make a direct comment on it? Here are some specific thoughts about this:

- What do your results suggest about the resolution of this debate?

- Can you relate greenness to GPP in the context of your findings of VI correlation with TWS, LST and VPD, or not?

- If not, do your results suggest something about leaf demography (drop, flush) and how it relates to drought?

- Are there field studies of leaf drop/flush to support an attempt to explain regional behavior?

- Can you relate your conclusions to other studies, such as those that look at SIF in the region (Doughty, Koren)?

**Response:** During the revision, we applied our 'exceeding normal ranges'-based method developed here to the Amazon drought in 2005 and 2010 and found that our method could resolve the decade-long debate. Below are the new figure and associated text in the revised manuscript.

'Our 'exceeding normal ranges'-based method developed herein can help resolve the debate around greenness anomalies in the dry season (July–September) of the 2005 drought (Saleska et al., 2007; Samanta et al., 2010). When we examined the MODIS-observed VI anomalies from May to October over the southern Amazon, both 2005 and 2010 witnessed a two-stage process: positive VI anomalies followed by negative VI anomalies (Fig. 10a and d). According to our method, the number of grid cells 'exceeding normal ranges' was very low in May, June, and July of both years (Fig. 10b and e), which means VI anomalies were primarily driven by PAR anomalies (Fig. 10c and f). Therefore, positive VI anomalies were observed during these months, with the strongest positive VI anomalies found in May 2005. With the progress of droughts, more than 50% of southern Amazon was found 'exceeding normal ranges' in August, September, and October 2005, while this number was greater than 75% in 2010. Therefore, stronger negative VI anomalies were observed in August, September, and October 2010, irrespective of radiation anomalies. When calculating the average VI anomalies for the transition months from positive to negative VI anomalies (i.e. average over July to September), it is very likely to obtain positive VI anomalies in 2005 but negative VI anomalies in 2010. Our results suggest that examining the hydrological, thermal and radiation conditions from the onset to the termination of droughts will enable us to better understand the responses of the Amazon rainforest.'

[Figure]

**Figure 10.** Temporal patterns of (a) standardized anomalies in vegetation indices (VI), (b) percentage of rainforest 'exceeding normal ranges' according to Approach #2c, and (c) standardized anomalies in photosynthetically active radiation (PAR) from May to October in 2005 over southern Amazon. Panel (d-f) Same as panel (a-c), but for the year 2010.

We also compared our results with other studies that look at SIF in this region during the 2015/16 drought.

'The spatiotemporal patterns of canopy greenness anomaly during the 2015/2016 drought found herein agree well with other independent satellite- and field-based vegetation observations. From the perspective of satellite observations, Koren et al. (2018) used the newly developed satellite-based sun-induced fluorescence (SIF) product (2007–2016) to examine the impact of the 2015/2016 Amazon drought. Temporally, it was found that the regional mean SIF was below its climatological average at the beginning and end of the drought, but above the average in the first half of 2016. Spatially, the eastern part of Amazon experienced much larger reductions in SIF than the western part.'

**R2C3:** I see TRMM precipitation mentioned in section 2.1 and Table 1, but don't recall seeing mention of precipitation elsewhere, as TWS is the variable used. There are interesting questions around the use of TWS from GRACE and TRMM precip. Neither product is perfect. Precipitation may evaporate off leaves (if light), and if heavy may run off before infiltration. TWS will have a significant contribution from soil well below maximum rooting depth, and that soil water might be irrelevant to the analysis. Furthermore, there may be lags between precipitation and plant function, but these lags may be accounted for by using TWS. If only one precipitation/soil moisture metric is used, then the other need not be listed in Table 1. Additionally, the authors should explain the reasoning behind the choice of one 'wetness' product over the other.

**Response:** (1) TRMM precipitation was used to calculate the Maximum Climatological Water Deficit (MCWD) with the results presented in the Appendix (Fig. C1 below). The similarities and differences between MCWD and our method developed herein were discussed in the Discussion section.

[Figure]

**Figure C1.** The difference between MCWD during August 2015–July 2016 and the mean MCWD of non-drought years (2003–2016, excluding 2005, 2010, 2015 and 2016) over the 1° grid cells with more than 80% covered by 'evergreen broadleaf forests'. MCWD stands for maximum climatological water deficit, and its calculation can be found in Aragão et al. (2007). The monthly precipitation data used here is derived from TRMM (TRMM 3B43 v7, see Table 1).

(2) During the revision, we replaced TWS with soil water product from ERA5-Land and performed the same analysis. The comparison (see new Table 2 and Table 3 below) shows that the choice of 'wetness' product will not essentially change the results of this study, which further demonstrates the robustness of the 'exceeding normal ranges'-based method developed in this study.

**Table 2.** Spatial agreement (%) between predicted VI anomaly direction derived from different approaches and MODIS-observed VI anomaly direction. There are 390 one-degree grid cells over the Amazon with more than 80% covered by 'evergreen broadleaf forests' considered in these statistics.

| Period | Approach #1 (Using PAR) | Approach #2A | Approach #2B | Approach #2C |
|---|---|---|---|---|
| | | (Using TWS, LST and VPD first, then PAR) | | |
| Stage I (August–October 2015) | 39% | 67% | 54% | **72%** |
| Stage II (November 2015–February 2016) | 66% | 68% | 68% | **68%** |
| Stage III (March–June 2016) | 72% | 72% | 72% | **72%** |
| Stage IV (July 2016) | 44% | 59% | 69% | **71%** |

**Table 3.** Spatial agreement (%) between predicted VI anomaly direction derived from different approaches and MODIS-observed VI anomaly direction. Same as Table 2, but TWS was replaced by soil water.

| Period | Approach #1 | Approach #2A | Approach #2B | Approach #2C |
|---|---|---|---|---|
| | (Using PAR) | (Using Soil Water, LST, VPD first, then PAR) | | |
| Stage I (August–October 2015) | 39% | 69% | 67% | **71%** |
| Stage II (November 2015–February 2016) | 66% | 68% | 68% | **68%** |
| Stage III (March–June 2016) | 72% | 72% | 72% | **72%** |
| Stage IV (July 2016) | 44% | 58% | 60% | **64%** |

**R2C4:** The calculation of seasonal cycle (various ND-Ave values) was calculated using the notion of a calendar year (Jan-Dec), but the analysis of the drought event was performed over the time of a 'water year' from August-July. In fact, this second methodology makes more sense, as the change in the calendar year comes in the middle of the wet season. My recollection is that many Amazonian researchers perform calculations over the scale of a water year. Why wasn't the notion of water year used consistently? Does it change the results when compared to the calendar year calculation?

**Response:** Done. Following the referee's suggestion, we switched the definition of a 'year' from 'January to December' to 'August to July' for all relevant figures in the revised manuscript for consistency. This switch did not change our results, as when we calculated the average value of non-drought years (i.e. ND-Ave), we took the average of the same month of non-drought years. For example, $TWS_{ND\text{-}Ave}$ for August is the average of TWS values in August from all non-drought years. To clarify this point, we added a better explanation of how we derived ND-Ave for each month. Below is the new figure in the revised manuscript.

[Figure]

**Figure 3.** Example illustrating how to derive (1) non-drought years' average and (2) non-drought years' extreme values of TWS, LST and VPD using the 1° grid cell centered at 9.5°S, 69.5°W. Panel (a) shows how we derived the non-drought years' average and extreme TWS values. Taking August for example, each grey dot represents August TWS value from one non-drought year, and there are ten non-drought years (i.e. 2003 to 2016, but excluding 2005, 2010, 2015 and 2016). The average of these ten TWS values is considered as the non-drought years' average in August (i.e. $TWS_{ND\text{-}Ave}$ in August). Following the same process, we derived $TWS_{ND\text{-}Ave}$ for the other 11 months. The minimum value of 12 $TWS_{ND\text{-}Ave}$ was taken as the extreme TWS ($TWS_{Min}$); for this example grid cell, October's $TWS_{ND\text{-}Ave}$ was chosen as $TWS_{Min}$. Panels (b) and (c) show the same as (a), but for LST and VPD. The extreme values of LST and VPD are $LST_{Max}$ and $VPD_{Max}$, respectively, which were reached in September and August during non-drought years for this example grid cell.

**R2C5:** On a related note, the years 2005, 2010 (and for that matter 2015/2016) were not drought everywhere. Were these denoted drought years just because other publications have said so, or was there an actual calculation of the fraction of the target gridcells during the (calendar) year that met drought criteria, and these years had the largest area under drought? Or was it that the metrics used to define drought was most severe?

**Response:** We fully agree with the referee that the Amazon rainforest was not in drought everywhere during these drought events. Actually, the objective of our 'exceeding normal ranges'-based method developed here is to identify the drought-affected regions. According to our method, regions 'exceeding normal ranges' during different stages of the 2015/16 event were delineated (see Figure 7 below). The results showed a gradual southward shift of these regions: from the north-eastern Amazon in August–October 2015, to the north-central part in November 2015–February 2016 and finally to the southern Amazon in July 2016.

[Figure]

**Figure 7.** Spatial distribution of terrestrial water storage (TWS), land surface temperature (LST) and vapor pressure deficit (VPD) anomalies for four stages over the 1° grid cells with more than 80% covered by 'evergreen broadleaf forests'. Coloured grid cells denote TWS, LST and VPD values are 'beyond non-drought years' extreme values' (i.e. $TWS < TWS_{Min}$ or $LST > LST_{Max}$ or $VPD > VPD_{Max}$). Hatched grid cells mean they are statistically significant ($p < 0.05$) different from the same months of non-drought years.

**Specific Comments**

**R2C6:** I think the references in lines 214-217 are for figure 3.

**Response:** Yes, thanks for your careful review (and apologies for our mistake). This has been corrected in the revised manuscript.

**R2C7:** Line 227: Figure 4a shows TWS above non-drought values for the first 2 months, not 3.

**Response:** Yes, correct. We have changed 'first three months' to 'first two months'.

**R2C8:** Line 228: "During Stage III, only a small area with TWS<TWS$_{ND-Min}$ occurred in the north-east." This is confusing. Figure 4a shows TWS well below non-drought TWS, with a consistent amplitude, over the period from December 2015 through July 2016. The maps in Figure 5 seem to contradict this. Is it a difference between a few very dry gridcells and a lot of 'sort of' dry gridcells? Some explanation of this apparent discrepancy might be helpful.

**Response:** To enhance clarity, we introduced an explanatory figure (see Figure 4 below) in the Methods section, illustrating the distinction between 'non-drought years' average' and 'non-drought years' extreme values'. Basically, among these grid cells with TWS below the same months of non-drought years, only parts of them are lower than TWS$_{Min}$.

[Figure]

**Figure 4.** Example illustrating (1) the difference between 'significantly (p<0.05) different from the same months of non-drought years' and 'beyond non-drought years' extreme values', and (2) how to determine the hydrological and thermal conditions 'exceeding normal ranges' in Approach #2A, #2B and #2C, respectively. In panel (a), terrestrial water storage (TWS) values in the drought year are 'significantly (p<0.05) different from the same months of non-drought years' for six months (i.e. September, October, November, May, June and July), but 'beyond non-drought years' extreme values' (i.e. TWS<TWS$_{Min}$) for only three months (September, October and November). In panels (b) and (c) the same is shown for land surface temperature (LST) and vapor pressure deficit (VPD), respectively. The months marked as #2A in panel (c) are considered 'exceeding normal ranges' according to #2A. Same for #2B and #2C marks in panel (c).

**R2C9:** Lines 294-299. Are your results consistent with the findings of this study? Do you find that you don't see a depletion of TWS in the region of the field studies of Fontes? If you do see a TWS depletion, and Fontes says it doesn't matter, what does that suggest about your method?

**Response:** During the revision, we added one figure plotting the temporal patterns of TWS, LST and VPD over the 1° grid cell centered at 2.5°S, 60.5°W (Fonte's grid cell) from August to December 2015.

Below are the new figure and associated text in the revised manuscript.

'Fontes et al. (2018) found that leaf and xylem safety margins (LXSMs) of central Amazonian trees showed a sharp drop in the months with unusually high canopy temperature and *VPD* from August to December 2015. LXSMs were significantly negatively ($p < 0.05$) correlated with *VPD*, but not with soil water storage. Moreover, the high values of predawn leaf water potential from 2015 through 2017 suggested that soil water supply was not limiting during their study period. These results indicate that the atmospheric demand could be the main driver for decreasing plants' LXSMs. We examined the anomalies of TWS, LST and VPD over Fontes' grid cell for the same period (August to December 2015) (Fig. 9). Strong positive anomalies in LST and VPD agree with the field measurements in Fontes et al. (2018). Moreover, TWS from August to November 2015 was higher than in the same months of non-drought years, suggesting sufficient soil water was available during this period.'

[Figure]

**Figure 9.** Temporal patterns of terrestrial water storage (TWS), land surface temperature (LST) and vapor pressure deficit (VPD) anomalies during August to December 2015 for the 1° grid cell centered at 2.5°S, 60.5°W. Panel (a) shows TWS for each month from August to December 2015 as well as for the non-drought years' average (± standard deviation). Panels (b) and (c) are the same as (a), but for LST and VPD, respectively.

Thanks very much for your thought-provoking comments; addressing them has improved the quality of our revised manuscript. We've thanked you for your efforts in the Acknowledgements section.